# Immune Response to Respiratory Viral Infections

**DOI:** 10.3390/ijms25116178

**Published:** 2024-06-04

**Authors:** Antonella Gambadauro, Francesca Galletta, Alessandra Li Pomi, Sara Manti, Giovanni Piedimonte

**Affiliations:** 1Pediatric Unit, Department of Human Pathology in Adult and Developmental Age “Gaetano Barresi”, University of Messina, Via Consolare Valeria 1, 98124 Messina, Italy; gambadauroa92@gmail.com (A.G.); francygall.92@gmail.com (F.G.); alessandra.lipomi92@gmail.com (A.L.P.); 2Office for Research and Departments of Pediatrics, Biochemistry, and Molecular Biology, Tulane University, New Orleans, LA 70112, USA; gpiedimonte@tulane.edu

**Keywords:** immune response, respiratory viral infection, RSV, influenza, SARS-CoV-2, newborns, pregnant women

## Abstract

The respiratory system is constantly exposed to viral infections that are responsible for mild to severe diseases. In this narrative review, we focalized the attention on *respiratory syncytial virus* (RSV), influenza virus, and *severe acute respiratory syndrome-coronavirus-2* (SARS-CoV-2) infections, responsible for high morbidity and mortality in the last decades. We reviewed the human innate and adaptive immune responses in the airways following infection, focusing on a particular population: newborns and pregnant women. The recent Coronavirus disease-2019 (COVID-19) pandemic has highlighted how our interest in viral pathologies must not decrease. Furthermore, we must increase our knowledge of infection mechanisms to improve our future defense strategies.

## 1. Introduction

Respiratory viral infections are a persistent public health problem, causing global morbidity and mortality, particularly in infants and children [1,2]. Respiratory viruses typically have airway epithelial cells as their primary target. The infection may be limited to the upper respiratory tract (URT) or progress to the lower respiratory tract (LRT), also inducing bronchiolitis or pneumonia [3,4,5]. The balance between the immune response and the viral infection differs significantly with age, host defense mechanisms, and viral load and properties [6,7,8]. Host defense against viruses involves several components of the immune system, including mechanical barriers (i.e., mucosal surfaces) and innate and adaptive immune responses [9].

Numerous viruses can cause respiratory infections, most commonly *rhinoviruses*, *respiratory syncytial virus* (RSV), *influenza* and *parainfluenza* viruses, *coronaviruses*, *metapneumoviruses*, *adenoviruses*, and *enteroviruses* [10,11]. RSV, Influenza virus, and *severe acute respiratory syndrome-coronavirus-2* (SARS-CoV-2) have been responsible for a significant number of deaths [12,13,14] despite the multiple preventive and vaccination strategies created to limit the worldwide spread of these infections. Moreover, newborns and pregnant women seem more susceptible to being infected by these three viruses [15].

This review aimed to summarize high-quality studies on the immune response to respiratory viral infections. We focused our research on immunity to RSV, the influenza virus and SARS-CoV-2 infections, mainly in special populations, such as newborns and pregnant women. By combining the key terms “respiratory viral infections” OR “Respiratory Syncytial virus” OR “RSV” OR “Influenza virus” OR “Severe Acute Respiratory Syndrome-Coronavirus-2” OR “SARS-CoV-2” AND “immune response” AND “newborns” OR “neonates” OR “pregnant women” OR “pregnancy” in a computerized search of PubMed limited to the last 10 years, our purpose was to provide a comprehensive overview of the existing literature, based on a critical evaluation without standardized methodologies or statistical analyses.

## 2. The Airway Mucosal Surface

The airway mucosal surface is the second largest surface area of the human body and is the first line of respiratory defense against pathogens, including viruses [16]. This barrier comprises structural cells (i.e., epithelial, endothelial, and mesenchymal cells) in contact with an active visco-elastic gel matrix known as airway mucus [17].

The ciliated epithelial cells are pivotal components of mucociliary clearance, as they trap and remove different insults [18,19]. Moreover, epithelial cells secrete antimicrobial factors, such as surfactant, complement, mucins, and antimicrobial peptides (AMPs), that guarantee an active resource against infections [20,21]. In particular, surfactants are produced by type II airway epithelial cells and allow the clearance of multiple harmful particles on the lower airway surface, where the mucociliary clearance mechanism is absent. Among the four surfactant-associated proteins (SPs), SP-A and SP-D are opsonins and cooperate with the innate immune cells (such as macrophages and neutrophils) by binding the viral capsids to glycosylated viral-attachment proteins, preventing the entry of viruses in the host cells and allowing their neutralization by phagocytes [22,23]. Within the lung parenchyma, epithelial and endothelial cells form the alveoli, mediating gas exchange and extravasation of immune cells into the respiratory tissue during inflammation. Below the epithelial cells, mesenchymal cells constitute a basal layer in which the extracellular matrix (ECM) is produced and cooperates against pathogens during acute inflammation by producing pro-inflammatory cytokines and chemokines [20,24].

The mucus produced by goblet cells consists of a complex net of water, mucins, cytokines, complement factors, antimicrobial molecules, and secretory immunoglobulin (Ig) A [20,25]. Mucins (MUC5B and MUC5AC), high-molecular-weight glycoproteins, interact with allergens and pathogens that can be trapped within the mucus and easily removed by the movement of cilia of the epithelial cells [26,27]. Furthermore, the airway mucus is populated by a beneficial commensal microbiota, noted as the lung microbiota. This “host-compatible” commensal microbiota was recently identified in the lung, and studies have shown its role during respiratory viral infections and its interaction with the host immune system [28,29]. The lung microbiota composition can affect the severity of the viral infection and associated immune response [30,31]. Moreover, microbial metabolites can impact the activation of the local immune cells during inflammation [32].

In addition to these physical (structural cells) and chemical (mucus) barriers, several immune cells are associated with the airway mucosal surface, including alveolar macrophages (AMs), innate lymphoid cells (ILCs), dendritic cells (DCs), and tissue-resident lymphocytes [20,33]. Moreover, accumulations of lymphoid cells are organized in specific compartments along the mucosa of the respiratory tract, including the nasopharynx-associated lymphoid tissue (NALT) in the URT and the bronchus-associated lymphoid tissue (BALT) in the LRT. Both tissues initiate local immune responses against viruses and preserve the memory of the infection in the respiratory tract [16,34,35].

## 3. Respiratory Viral Infections and Immune Response

Respiratory viruses use different receptors to infect airway epithelial cells. Viruses can fuse their structure with the epithelial cell membrane through the distinctive binding to a cellular receptor or be endocytosed. After that, they enter the cytoplasm or nucleus via vesicle fusion or by translocating across the cell membrane or inducing the rupture of the endocytic vesicle once in the cytoplasm. Following the replication within cells, viruses induce cytolysis to facilitate the spread of the infection or bud from the plasma membrane of infected cells without inducing cellular death [9,36]. The viral binding to their receptors activates the immune response. The innate immune response is the first line of defense against general pathogens. It consists of physical, chemical, and cellular defenses, while the adaptive immune response is the second line of defense and is antigen-specific [3] (Figure 1).

### 3.1. Innate Immune Response

The innate immune response inhibits viral infection, protects cells against pathogens, and eliminates virus-infected cells [9]. In the beginning, specific molecular signatures expressed on the cell surfaces or cytoplasm noted as pathogen-associated molecular patterns (PAMPs) are detected by pattern recognition receptors (PRRs) expressed by epithelial and immune cells in the airways [20]. PAMPs are evolutionarily conserved structures localized on pathogens the host recognizes as “not-self” [37]. This recognition involves PRRs, a class of receptors classifying into five types based on protein domain homology: toll-like receptors (TLRs), nucleotide oligomerization domain (NOD)-like receptors (NLRs), retinoic acid-inducible gene-I (RIG-I)-like receptors (RLRs), C-type lectin receptors (CLRs), and interferon gene stimulators (STING) [16,38]. Some of these receptors are specifically implicated in respiratory viral recognition; for instance, among TLRs, TLR3, TLR7/8, TLR9, and TLR10 are expressed on the cell membrane, while among RLRs, RIG-I, MAD5, and LGP2 are soluble PRRs localized in intracellular compartment membranes and cytoplasm [38,39]. The activation of PRRs by their specific ligands promotes several signaling pathways (i.e., nuclear factor-kB (NF-kB) signaling, mitogen-activated protein kinase (MAPK) signaling, TANK-binding kinase 1 (TBK-1)–interferon regulatory factor 3 (IRF3) signaling, and inflammasome signaling), which are responsible for inducing immunoprotective effects and connecting innate and adaptive immune responses [38]. The primary function of these signaling cascades is to produce and secrete numerous cytokines, chemokines, and interferons (IFNs). IFNs are particularly well-known for their antiviral properties [39].

IFNs are a group of signaling proteins secreted by various host cells in response to pathogens, including viruses. There are three families of IFNs: type I, type II, and type III, each with distinct sequence homology, receptors, and functional activities. Type I consists of multiple IFN-α subtypes and single IFN-ß, IFN-ε, IFN-κ, and IFN-ω, all signaling through the IFN alpha receptor (IFNAR). Type II comprises a single member, IFN-γ, and its signaling occurs through the IFN gamma receptor (IFNGR). Type III is represented by four subtypes, IFN-λ1 (IL-29), IFN-λ2 (IL-28A), IFN-λ3 (IL28B), and IFN-λ4, all signaling through the IFN lambda receptor (IFNLR). Type III IFNs are the first line of defense against viral replication in epithelial cells in mucosal barriers and control the infection locally rather than systemically. Type I IFNs have more potent inflammatory properties and are produced by various cells, including airway epithelial cells, AMs, monocytes, and plasmacytoid DCs (pDCs). The binding between IFNs and their receptors activates the Janus kinase (JAK) signal transducers and activators of the transcription (STAT) pathway, resulting in the transcription of IFN-stimulated genes (ISGs) [16,39,40]. ISGs have direct antiviral effects because they detect viral RNA and inhibit viral replication. Moreover, ISGs can enhance the IFN signaling pathways and present a paracrine function, limiting the spread of viral infection to adjacent cells. However, viruses have developed various strategies for limiting the activity of the JAK–STAT pathway [16,40].

Among the several immune cells associated with the airway mucosal surface, AMs are the most numerous pulmonary leukocyte population located in alveolar spaces, where they play a role in facilitating gas exchange, regulating surfactant catabolism, and guaranteeing innate immune response [3,20]. AMs are lung-resident sentinel cells [41] and exist typically in a semi-quiescent state: they express numerous PRRs for detecting “not-self” signals and once activated, produce pro-inflammatory cytokines and growth factors (such as interleukin (IL)-6, tumor necrosis factor (TNF)-α, CCL2, and granulocyte-colony stimulating factor (G-CSF)) by which they promote the host innate response against pathogens [20]. Recent studies have underlined the central role of AMs during viral infections. AMs recognize the pathogens, start their production of pro-inflammatory cytokines, and recruit other inflammatory cells, increasing lung protection against viral infection. Moreover, AMs are the main producers of type I IFNs during viral infection, which is noted to suppress early viral replication [42]. As tissue damage accumulates, AMs remove apoptotic cells from the respiratory surfaces by phagocytosis, maintaining lung homeostasis [20].

During respiratory viral infections, neutrophils are particularly abundant in the air- ways. They limit viral replication and spread by phagocytosis and cell apoptosis [3]. Furthermore, neutrophils release toxic proteins and enzymes, including myeloperoxidase (MPO), elastase, and defensins, that improve the immune response to extracellular pathogens but can damage surrounding tissues. A third mechanism of neutrophils’ action consists of the formation of neutrophil extracellular traps (NETs), extracellular networks of DNA fibers studded with several granule proteins (e.g., MPO, elastase), and nuclear proteins (e.g., histones). Even in this case, however, the balance between aid and damage to the host is delicate, with recent research showing that NETs can have a direct cytotoxic effect on the lung epithelium, increasing the risk of developing future airway obstruction [43].

During respiratory viral infections, eosinophils can participate in the innate immune response directly and indirectly by producing some eosinophil-released products with antiviral properties. Among these products, eosinophil-derived neurotoxin (EDN) and eosinophil cationic protein (ECP) stimulate the activation of the adaptive immune response against viruses. Moreover, eosinophils express different PRRs (such as TLR3, TLR7, and TLR9) that recognize viruses and activate these granulocytes to produce various cytokines and chemokines (i.e., IL-6, IP10, CCL2, and CCL3) that increase viral clearance [44]. Similarly to neutrophils, eosinophils can create eosinophil extracellular traps (EETs), networks of DNA fibers and several granule proteins (e.g., major basic protein (MBP) and ECP) that take part in the innate immune response against multiple infections [45].

Natural killer (NK) cells, inflammatory monocytes, unconventional T cells, and DCs increase in number during viral infection in the airways. NKs kill virus-infected cells by releasing cytotoxic granules and activating death receptors. By using these cytotoxic processes, NKs limit viral replication and spread. Moreover, they provide an early source of IFN-γ and activate the adaptive immune response by stimulating T cells. DCs are typically localized in the airway mucosal surface near the airway epithelial cells and within alveoli walls. DCs can be activated by viruses through PPRs and pro-inflammatory cytokines and chemokines released by airway epithelial cells and other resident immune cells [46,47]. pDCs are pivotal producers of IFN-α during respiratory viral infections and play a role in starting the innate immune response. Conventional DCs are primarily associated with the mucosal epithelium of the conducting airways and have an important role in stimulating adaptive immune response by presenting viral antigens to CD4 and CD8 naïve and memory T cells [39,46,48].

### 3.2. Adaptive Immune Response

Adaptive immunity performs an important line of defense against respiratory viral infections. While the innate immune response is rapid, taking minutes to hours following the detection of PAMPs, the adaptive immune response is antigen-specific. It takes several days to weeks to act [20]. Adaptive immunity is essential for eliminating viral infections and establishing a long-lasting immunological memory against a specific pathogen that acts faster in eliminating subsequent insults [9,16]. The adaptive immune response involves three main cell types: B cells, responsible for the production of antibodies; T CD4 cells, which include different subtypes (Th1, Th2, Th17, T-regulatory cells (Treg), and T-follicular helper cells (Tfh)); and T CD8 cells, which are responsible for the direct killing of infected cells in the airways [16,20]. The adaptive immune response starts in the draining lymph nodes localized in the NALT and BALT, where the antigen-presenting cells (APCs), primarily conventional DCs, migrate after their activation by respiratory viruses.

T CD4 and CD8 naïve cells are the first to be activated by migratory DCs during respiratory viral infections. After their activation, these cells become effectors and can migrate to the infectious site to orchestrate the antiviral immune response. T CD8 cells move to the respiratory tract and mediate the targeted lysis of infected cells through the secretion of cytolytic enzymes (e.g., granzyme B and perforin). Another mechanism by which the infected cells are eliminated is the direct cell-to-cell contact through the interactions of surface molecules such as Fas (CD95) and FasL (CD95L) or through the TNF-related apoptosis-inducing ligand (TRAIL) expressed on T CD8 cells. Moreover, T CD8 cells produce pro-inflammatory cytokines following a respiratory viral infection, such as IFN-γ and TNF, which can promote infected cell death. Pro-inflammatory cytokines seem to enhance the production of the regulatory cytokine IL-10, secreted by both T CD8 and Treg cells, which modulates the inflammatory response [49,50,51]. Among the different subtypes of T CD4 cells, Th1 cells are potent during the acute respiratory infection, promoting the activation of resident macrophages and producing numerous cytokines (such as IFNγ and IL-2); Th2 cells regulate inflammation in the airways through multiple cytokines, including IL-4, IL-5, IL-9, IL-10, and IL-13, and are involved in the production of antibodies; Th17 cells increase the activity of Th2 cells, stimulate lung neutrophilic infiltration and modulate the responses of T CD8 cells; Treg cells produce IL-10 and transforming growth factor (TGF)-β and are responsible for maintaining homeostasis during the acute infection, also by promoting tissue repair; and Th cells promote humoral response and facilitate the B cells’ class switching [20,39,52]. A new subtype of T CD4 cells, termed T resident helper (TRH), was recently described and operates locally by promoting humoral response in the lung [53]. During the later stages of the immune response, virus-specific memory T cells develop and persist over decades within various T cell compartments as sentinels against reinfection [54]. However, a subset of T memory cells, noted as tissue-resident memory T (TRM) cells, remain in the respiratory tract throughout life and protect the lung against reinfection [55].

During primary respiratory viral infection, B cells are activated by APCs (especially DCs) in the lymphoid tissues (NALT and BALT). Thus, the subsequent interaction with T CD4 helper cells leads to immunoglobulin isotype class switching from IgM to IgG and IgA [56]. Both IgA and IgG neutralize respiratory pathogens through opsono-phagocytosis and promote viral clearance in the airways. In the later stages of infection, subsets of B cells differentiate in B memory cells and tissue-resident memory B (BRM) cells, guaranteeing a prompt immune response in subsequent viral infections [20,57].

## 4. Host Responses to RSV, Influenza Virus, and SARS-CoV-2

Viral respiratory infections constitute a prominent cause of global mortality, and LRT infections (LRTI) are the fourth leading cause of mortality worldwide [20]. RSV, influenza virus, and SARS-CoV-2 have peculiar replication cycles, cellular targets, and host response evasion mechanisms (Table 1). Moreover, they can cause a wide range of respiratory diseases, from mild rhinitis to pneumonia [16]. Understanding the infectious mechanisms and specific host immune responses is crucial for reducing the spread of these diseases and improving therapeutic management.

### 4.1. Innate and Adaptive Immune Response to RSV

RSV infection frequently leads to LRTI in infants and young children, exhibiting a spectrum of severity from mild symptoms like rhinitis to more severe conditions such as bronchiolitis and pneumonia [58,83]. Throughout an individual’s life, reinfections with RSV are expected, potentially heightening respiratory issues like recurrent wheezing and asthma due to inadequate establishment of immunological memory [84]. The innate immune system serves as a crucial first line of defense against RSV infections, with various components, including diverse types of innate cells, PRRs, and numerous cytokines and chemokines working collaboratively against RSV infection. Generally, the innate cellular immune response against viral infections is primarily mediated by type I IFNs, which trigger the expression of numerous ISGs with diverse effects [85].

In nasopharyngeal samples from RSV-infected children, an IFN response signature is often observed despite most patients’ absence of detectable secreted IFNs at the transcriptional level. These findings indicate that the nasal mucosa may not be the primary source of secreted IFNs in these infections [16]. RSV is known to induce IFNs in the respiratory tract weakly. Infants infected with RSV exhibit increased activation of IFN-related genes in peripheral blood samples 4–6 weeks post-infection compared to the acute phase, suggesting potential suppression of the IFN response during acute RSV infection. Specific RSV proteins, such as G and F proteins, have been implicated in inhibiting IFN production and exacerbating viral load and mucus production [65]. However, levels of IFN-λ 1–3 have been associated with the severity of respiratory disease caused by RSV, suggesting a complex role for IFNs in RSV infection [66]. Several ISGs, such as ISG15, ISG56 (IFIT1), MX1, and OAS1, play crucial roles in inhibiting RSV replication through various mechanisms and contribute to shaping the adaptive immune response to RSV [86].

Additionally, distinct viral proteins/genomes and replication products from RSV act as intrinsic PAMPs, initiating the host’s innate immune response via PRRs [58,68]. So far, three primary classes of PRRs—TLRs, RLRs, and NLRs—have been implicated in recognizing RSV [68]. TLRs are constitutively expressed on various cell types and interact with RSV, triggering downstream signaling pathways essential for controlling viral replication and modulating the adaptive immune response [58,68]. Specifically, several TLRs, such as TLR2, TLR3, TLR4, TLR7, and TLR8, have been implicated in RSV recognition. For instance, TLR3 detects double-stranded RNA (dsRNA), a common intermediate in RSV replication. At the same time, TLR4 can recognize RSV’s structural proteins, like the F protein, activating NF-κB-mediated cytokine responses [58]. Leukocytes expressing TLR2, TLR3, TLR4, TLR6, and TLR7 also interact with RSV, promoting immune responses post-infection [87]. RIG-I and melanoma differentiation-associated protein 5 (MDA5) are the primary RLRs that recognize viral RNA intermediates generated during RSV replication [59]. Despite the higher baseline RIG-I levels in children’s respiratory tracts and its significance in RSV control, they remain highly susceptible to RSV infection, suggesting effective viral escape mechanisms inhibiting RIG-I in the respiratory tract [16]. Several NLRs, including NLRP3 and NLRC5, have been implicated in the response to RSV infection, with NLRP3 activation promoting the maturation and secretion of pro-inflammatory cytokines and NLRC5 regulating the expression of IFN-I [58,60]. The adaptive immune response to RSV involves the coordinated activation of T and B cells, which are crucial for eliminating the virus and providing long-term immunity. Upon initial exposure to RSV, DCs, acting as APCs, present viral antigens to T cells in the draining lymph nodes. This process initiates the activation and proliferation of RSV-specific CD4+ T helper cells, which subsequently drive the differentiation of cytotoxic CD8+ T cells and activate B cells [68]. Notably, RSV infection triggers a robust memory B cell response in the adenoids of young children, and nasal antibody titers correlate with protection against RSV [88]. B cells produce antibodies targeting RSV proteins, with anti-F protein antibodies from young children’s adenoids exhibiting superior neutralization capabilities [72]. Neonatal B cells, infected by RSV via CX3CR1, contribute to heightened pathology and a Th2 response. Recent findings indicate a link between the B cell response to RSV and the expression of IFN-I receptors, potentially resulting in decreased B cell function in newborns due to reduced IFN responses induced by RSV [73].

The Th2 response, characterized by the activation of type 2 helper T lymphocytes, plays a significant role in RSV infections, contributing to antibody and eosinophil responses. An imbalance in the Th1 response can lead to severe lung complications and heightened inflammation, particularly in children [69]. Different T cell profiles, including CD8 T cells expressing IFN-γ (Tc1) and IL-17 (Tc17) associated with shorter hospitalization and Th17 cells, have been linked to varying disease severity. Infants infected with RSV during their first year of life exhibit diminished antiviral memory T cell responses by the age of 2–3 years compared to uninfected counterparts [89].

RSV employs various strategies to evade the host immune response and establish infection [68]. Key players like NS1 and NS2 suppress IFN-I production and signaling, diminishing the host’s ability to mount an effective immune defense against the virus [77]. Additionally, through G, N, M, and SH proteins, RSV disrupts innate immune recognition, interfering with the recognition of viral components by PRRs, and modulates the host’s innate immune response through various mechanisms, ultimately facilitating persistent infection and recurrent respiratory tract infections, especially in vulnerable populations such as infants and the elderly [78].

### 4.2. Innate and Adaptive Immune Response to SARS-CoV-2

The SARS-CoV-2 virus, a member of the Coronaviridae family, is responsible for the coronavirus disease (COVID-19), which has had a significant global impact since the pandemic’s emergence in 2019 [90]. The replication cycle of SARS-CoV-2 begins with the virus binding to the ACE2 receptor via the S1 subunit of the spike protein. This interaction triggers a conformational change in the spike protein, facilitating the fusion of the viral and cellular membranes through the S2 subunit [67]. SARS-CoV-2 primarily targets the URT, initiating the mucosal immune response in areas such as the nasopharynx, including tonsils and adenoids [90]. IgA plays a crucial role in mucosal immunity, effectively combating viral replication and reducing the risk of reinfection. Recent studies have shown that even some seronegative patients with mild COVID-19 have detectable IgA with neutralizing activity in mucosal sites. Interestingly, IgA levels tend to increase with age, peaking in adolescence, but elevated levels in adult patients have been associated with worse prognosis and increased fatality rates [91,92]. However, IgA levels remain elevated even in the convalescent phase in patients with multisystem inflammatory syndrome in children (MIS-C), indicating a sustained mucosal immune response [93].

Upon infection with SARS-CoV-2, the host initiates a robust innate immune response mediated by PRRs, such as RIG-I, MDA-5, and TLRs. TLR2 detects the viral envelope protein, triggering the release of TNF-α and IFN-γ, while TLR3 responds to viral PAMPs and dsRNA, inducing the production of IL-1β and IL-18 via the NLRP3 inflammasome [61]. TLR4, TLR1, and TLR5 are predicted to respond to the SARS-CoV-2 spike glycoprotein, activating innate immunity signaling pathways. Endosomal receptors TLR7, TLR8, and TLR9 detect viral RNA, realizing various cytokines, including IFN type I and III [62]. Activation of RIG-I and MDA5 by dsRNA results in the phosphorylation of IRF3, initiating NF-kB signaling and stimulating cytokine production, such as IFN type I and III [63]. IFN I and III bind to their respective receptors, activating the JAK/STAT pathway and inducing the expression of MHC class I and ISGs, which inhibit viral replication. Suppressed IFN response is associated with COVID-19 severity. Viral proteins are degraded by proteasomes and presented on MHC class I proteins for cytotoxic T cell destruction of infected cells [67]. Dysregulation of cytokine release contributes to tissue and organ damage, exacerbated by inflammatory cell migration to the lungs [61].

Subsequently, the host mounts an adaptive immune response involving a complex interplay of immune cells and molecules, crucial for recognizing and eliminating the virus. DCs present viral antigens to T cells, activating and increasing virus-specific CD4+ T helper cells [74]. These CD4+ T cells play a central role in coordinating the immune response by providing help to B cells for antibody production and activating CD8+ cytotoxic T cells [70]. B cells producing neutralizing antibodies against SARS-CoV-2 provide a vital defense mechanism against viral infection [74]. Memory B cells are also generated to confer long-term immunity, ensuring a rapid and effective response upon subsequent exposure to the virus [70,74]. However, acute COVID-19 is marked by the absence of germinal centers, leading to the generation of “disease-related” B cells with limited protective capacity [75]. CD8+ cytotoxic T cells target and eliminate virus-infected cells, preventing the spread of the virus throughout the body [70].

Similarly to RSV, SARS-CoV-2 adopts several evasion mechanisms to evade host immune responses, enabling it to establish infection and replicate efficiently. In this regard, SARS-CoV-2 inhibits the production and signaling of IFNs, escaping early immune detection and delaying the activation of antiviral defenses. Additionally, SARS-CoV-2 can evade recognition by TLRs and RLRs and modulate antigen presentation, interfering immune cells with the ability to recognize and respond to infected cells. Furthermore, SARS-CoV-2 can manipulate cytokine signaling pathways, leading to dysregulated immune responses and exacerbated inflammation, contributing to disease severity. Finally, the virus can undergo antigenic variation, evading recognition by pre-existing immunity and potentially leading to reinfection or reduced vaccine efficacy [79].

### 4.3. Innate and Adaptive Immune Response to Influenza Virus

Two types of influenza viruses cause significant disease in humans: influenza A and influenza B. Influenza A virus includes subtypes A (H3N2) and A (H1N1), while influenza B virus is classified into B/Yamagata/16/88-like (B/Yam) and B/Victoria/2/87-like (B/Vic) lineages. Infection with these viruses can result in mild respiratory symptoms primarily affecting the URT, such as fever, sore throat, rhinitis, cough, lethargy, and headache. However, more severe cases can lead to LRTIs, potentially resulting in viral or bacterial-induced pneumonia, which is particularly dangerous for older adults [94]. Occasionally, global pandemics of the influenza A virus to arise due to introducing a new strain different from the circulating viruses, which generally come from animal reservoirs [95,96]. Lack of immunity to the new strain leads to increased morbidity and mortality, affecting various demographics, not just high-risk groups [96].

All influenza viruses have an enveloped negative-sense single-stranded RNA genome composed of eight segments encoding essential proteins like RNA polymerase subunits, viral glycoproteins, such as hemagglutinin (HA), and neuraminidase (NA), viral nucleoprotein, matrix protein, membrane protein, and nonstructural protein. HA facilitates viral entry into host cells, while NA aids viral release [95]. Upon infection, influenza viruses trigger innate immune responses mediated by PRRs, such as TLR3, TLR7, TLR8, cytosolic receptor RIG-I, and NOD-like receptor NLRP3. Activation of these pathways induces the expression of IFNs (IFN-I/III) and pro-inflammatory cytokines, which stimulate antiviral ISGs and recruit immune cells. The NLRP3 inflammasome releases cytokines (IL-1β, IL-18), triggering pyroptosis in infected cells. Knockout animal models demonstrated the protective role of these pathways, with ISGs, like Mx and IFITMs, exhibiting antiviral activity by inhibiting viral replication [64]. Early and robust innate immune activation correlates with host protection against the influenza virus in animal models [97]. Macrophages secrete cytokines, initiate adaptive immunity, and clear debris [98]. AMs in lung tissue play a crucial role in influenza defense, but excessive cytokine activation contributes to lung injury [97,98]. Wnt/β-catenin/HIF-1α and PPAR-γ signaling promote inflammatory activity in AMs, with PPAR-γ downregulation critical for controlling inflammation [97,99].

The adaptive immune response to the influenza virus involves the activation of T and B lymphocytes, leading to antigen-specific memory cells that confer long-term immunity [71]. Upon initial exposure to the virus, dendritic cells present viral antigens to naïve T cells in the lymph nodes, triggering their activation and differentiation into effector T cytotoxic CD8+ T cells that play a critical role in the clearance of the influenza virus and subsequent host recovery through the production of antiviral cytokines and direct killing of infected cells. These cells recognize short peptide fragments from conserved viral proteins presented by MHC class-I molecules, providing broad cross-reactive immunity [100]. Animal studies have demonstrated the importance of CD8+ T cells in cross-protection against different influenza subtypes and accelerated viral clearance upon reinfection [71,101]. In humans, CD8+ T cell responses have been associated with efficient virus clearance and reduced disease severity during influenza infections. Individuals with pre-existing CD8+ T cell responses showed lower disease severity during seasonal epidemics and the 2009 H1N1 pandemic [102]. CD4+ T cells recognize viral peptide fragments presented on major histocompatibility complex class-II (MHC-II) molecules and play a crucial role in enhancing CD8+ T cell and B cell responses during influenza infection. Activated CD4+ T cells produce cytokines, such as IFNs and IL-2, and migrate to the lungs to aid viral clearance. Circulating Tfh cells are a prominent CD4+ T cell subset identified during influenza infection and vaccination, providing help to memory B cells for antibody production. CD4+ T cells also assist in licensing antigen-presenting dendritic cells and contribute to the generation and recall of CD8+ T cell memory [71].

Furthermore, they can acquire cytotoxic functions, particularly in mucosal tissues like the lungs, further aiding in viral clearance [81]. B cells undergo clonal expansion and differentiation into plasma cells, secreting virus-specific antibodies, primarily targeting the surface glycoproteins HA and NA. These antibodies neutralize viral particles, inhibit viral entry and release, and promote opsonization for phagocytosis. Memory T and B cells are also generated, providing rapid and robust responses upon subsequent encounters with the virus [76].

Influenza viruses have evolved various evasion mechanisms to evade host immune responses and establish successful infection. One such mechanism involves the rapid mutation of HA and NA, which allows the virus to escape recognition by pre-existing antibodies and evade neutralization [80]. Additionally, influenza viruses can undergo antigenic drift and shift, leading to the emergence of novel strains with altered antigenic properties, further complicating immune recognition [81]. Another evasion strategy involves inhibiting host antiviral pathways, such as the IFN response, by viral proteins like NS1, which suppresses the production of IFN and other antiviral cytokines, allowing the virus to replicate unchecked [82]. Furthermore, influenza viruses can modulate host immune responses by inducing immunosuppression or dysregulation, impairing immune cells’ activation and function, thereby facilitating viral persistence and dissemination. These evasion mechanisms collectively contribute to the ability of influenza viruses to evade host immune surveillance and cause recurrent infections, posing significant challenges for vaccine development and antiviral therapy [80].

## 5. Special Population: Pregnant Women and Newborns

During pregnancy, the maternal immune system undergoes significant changes to avoid rejecting the developing fetus, which, together with the placenta, expresses paternal antigens that are “not-self” to the mother [15,103]. In the absence of immunomodulation, activation of the maternal immune system against a semi-allogeneic fetus could have catastrophic consequences, such as pregnancy loss [104]. In this context, sex hormones are essential in promoting maternal–fetal immunomodulation at the expense of antiviral immunity [15].

First, trophoblast cells surrounding a growing embryo and placenta after the implantation produce human chorionic gonadotropin (hCG), the hormone for the maternal recognition of pregnancy. HCG promotes the maintenance of the corpus luteum, which secretes progesterone during the first trimester [104]. HCG and estriol, not present in nonpregnant women, increase Treg expansion and enhance their function in suppressing effector T cells and other immune cells, promoting their role in providing self-tolerance [105,106]. During implantation, there is a shift from a Th1 profile, which controls trophoblast adhesion and stimulates the release of prostaglandin, to a Th2 profile. The Th2 profile promotes a largely humoral immune response and moderately suppresses the maternal immune response. This allows the maternal immune system to tolerate a semi-allogeneic fetus while maintaining vigilance [104]. Th2 dominance is less adept at counteracting viral infection. For this reason, pregnant women are more susceptible to viral respiratory diseases, such as the influenza virus, RSV, and SARS-CoV-2 [15,104,107]. Moreover, progesterone takes part in this process by stimulating the synthesis of a progesterone-induced binding factor that promotes CD4+ T cell/Th2 differentiation, with increased serum concentrations of Th2 cytokines, including IL-4, IL-5, and IL-10 [108,109].

Also, B cells change during pregnancy under the influence of estrogens, which can up- and downregulate their immunological function. If this does not occur, circulating maternal B cells would be recognized as partially foreign paternal human leukocyte antigen (HLA) genes the fetus expresses. Moreover, regulatory B cells (Breg) suppress the pro-inflammatory maternal immune response by secreting IL-10, which is also upregulated by Th2 cells. On the other hand, B cells protect from infectious pathogens by producing natural antibodies to supplement the maternal immune response, which can cross the placenta and strengthen the offspring’s immune system [104,110]. Specifically, B cells are responsible for fighting against most of the viral infections, such as SARS-CoV-2. In pregnant women, estrogen and progesterone reduce lymphopoiesis, so they have lower antibody titers against the virus [104].

Ultimately, ILCs, specifically NK cells, which play a critical role in viral clearance, are reduced during pregnancy, and TLRs lose the pattern recognition function [15,111].

### 5.1. Role of RSV Infection in Pregnant Women and Newborns

There are limited data about the immune response against RSV during pregnancy, but efforts are being made to encourage mothers to get vaccinated to protect their infants. Ma et al. [112], in their systematic review and meta-analysis, have shown that vaccination of the mothers results in an elevation of their antibody levels, which are transferred to the fetus through the placenta, giving passive immunity to the newborn, with greater efficacy to prevent LRTIs within the first 150 days after birth. There needs to be more knowledge about the immune response to RSV in newborns because of ethical and practical problems. Much information comes from animals, adults, or severe infections with complications, such as bacterial superinfection [113,114].

First, the innate immune system is essential in responding to the virus and its out- comes. The viral detection in the respiratory tract starts with the epithelial cells and fibroblasts (which express PPRs, such as TLR4 and TLR3) and APCs. After their ligation, an upregulation of cytokines (IL-6, TNF-α) and chemokines (CXCL10, CXCL8, CCL2, CCL3, CCL5) occurs [115], together with IFN (Figure 2). In this context, RSV can subvert viral clearance by inhibiting IFN-α/β signaling via its NS1 and two proteins [99]. pDCs are potent producers of IFN-α/β and are mobilized to nasal mucosa during RSV infection [100]. They are helped by NKs that provide an early IFN-γ source [47,116]. Moreover, neutrophils are recruited to the upper and lower airways [113]. Moving on to adaptive immunity, T cell response is fundamental to help antibody production and promote viral clearance. As said previously, most of the studies are performed on mouse models, showing that after infection, a peak of lymphocyte CD8+ in the lungs is present [113]. Furthermore, studies on infants have reported elevated IL-17, the Th17 promoting cytokines (IL-1β, IL-6, and IL-23), and CD161+ Th17 in the tracheal aspirate of RSV-infected infants, and IL-17+ CD4+ cells elevated in peripheral blood [117].

Despite palivizumab prophylaxis efficiently conferring protection to the lower airway when titer is sufficient, and antibodies having high affinity, this does not happen in the case of natural infection, in which the antibodies produced are poorly protective against reinfection [113].

### 5.2. Role of Influenza Virus Infection in Pregnant Women and Newborns

During pregnancy, viral infections, like influenza, can cause complications. Neutrophils, macrophages, and DCs support the immune response against the influenza virus. DCs recognize the virus through PRRs and induce the antiviral response by promoting CD4+ and CD8+ T cell immunity [118].

TLRs present in macrophages and DCs recognize pathogens through PAMPs, activating the immune response and the release of cytokines. A vital role in the immune reaction of respiratory epithelial cells to the influenza virus in humans is played by TLR3, which regulates the transcription factors for IFN, NF-kB, and MAPK through a pathway that involves myeloid differentiation factor 88 (MyD88) [119,120]. Furthermore, TLRs expressed on pDCs, when recognizing viral components such as genomic DNA and RNA, take part in the secretion of IFNs [118].

Different types of IFNs play a role in the immune response. Type-1 IFN helps in the maturation of DCs and induces the production of chemokines. These chemokines recruit monocytes and lymphocytes to the inflamed sites [109,121].

An important role has been recognized in TNF-a, which simultaneously limits viral replication and reduces viral-induced tissue damage, supporting the repair process (Figure 2) [109,118].

The role of adaptive immunity can be “homotypic” if the woman has already been exposed to the same serotype and depends on circulating IgG or “heterotopic”, which derives from previous infections with a different influenza serotype. In the latter case, influenza-specific memory CD4+ T cells can help the optimal B and T cell response. Recent pandemic strain outbreaks have demonstrated the critical role of vaccination in protection against severe disease [122].

The infections can modify the immune system during pregnancy, as has been demonstrated for Ig subclasses. In detail, a study of a cohort of pregnant patients infected with the H1N1 pandemic influenza in 2009 showed a reduction in IgG2 levels in the infected group compared to healthy pregnant controls. For this reason, H1N1 infection was more severe in pregnant women [123]. This was confirmed by a Chinese study, showing an imbalance of Ig subclasses against H1N1 and dysregulation in cytokines when compared to nonpregnant women of uninfected pregnant controls [124].

Together with tetanus–diphtheria–pertussis (Tdap), vaccination against the influenza virus using inactivated influenza vaccines (IIV) is recommended during pregnancy. Possible influenza infection and other viral respiratory infections, such as RSV and rhinovirus, can contribute to the overall beneficial response in the newborn regarding lower hospitalization rate and milder disease course [125].

The transfer of maternal antibodies can occur transplacentally and with breast milk. The first mechanism is an active transport process that involves pathogen-specific maternal antibodies being transported to the fetus through the placental Fc-receptor that binds to the mother’s IgG antibodies [126]. The likelihood of transmission of maternal antibodies across the placenta increases with an increase in their levels [127].

Breast milk is a valuable source of secretory IgA, which helps protect the baby against infections. IgA antibodies are produced in the mammary gland and transported to the breast milk via transcytosis. In this way, influenza-specific and neutralizing IgA are transferred to the neonate upon influenza vaccination of the pregnant mother [128]. This is associated with fewer respiratory diseases during the infant’s first six months [125].

### 5.3. Role of SARS-CoV-2 Infection in Pregnant Women and Newborns

The peculiarities of pregnant women’s immune response against SARS-CoV-2 and the effects of maternal immunization on the offspring are still topics on which research has focused during the last few years.

Several studies have evaluated the humoral response in pregnant women with SARS-CoV-2, highlighting possible differences in antibody response. An American study compared different types of IgG and IgM levels among pregnant and nonpregnant women following SARS-CoV-2 infection, showing no significant differences between the two groups [129]. Similarly, Kubiak et al. [130] have characterized the serologic response of pregnant women as correlating with the severity of clinical presentation and with the passive immunity to the neonate, finding higher antibody levels in symptomatic pregnant women compared to asymptomatic ones. Moreover, the levels of neonatal IgG positively correlated with maternal levels.

Moreover, research has focused on the possible specific effect of SARS-CoV-2 infection on maternal and fetal immunity. In detail, no traces of SARS-CoV-2 have been found in placenta samples. In umbilical cord blood (UCB) samples, IgM anti-SARS-CoV-2 were undetectable, whereas IgG antibodies, IL-1β, IL-6, and IFN-γ, were found in all cases (Figure 2). Moreover, a significant correlation was found between CD3+ mononuclear cells, CD3+/CD4+, and CD3+/CD8+ T cell subsets in the fetus and the concentration of IgG in UCB, showing a substantial immune response transmitted to the fetus in the case of maternal infection [131].

There is some positive news from an Italian study that SARS-CoV-2 infection did not cause significant complications during pregnancy or for newborns, whether the women were asymptomatic or symptomatic. Even women with pneumonia who received prompt clinical care did not experience significant complications. The study also looked at the different antibody trends in the two groups. They found that women with pneumonia had significantly higher levels of anti-S IgG, IgA, and anti-NCP IgG between 1 and 3 months after the onset of infection compared to asymptomatic women. On the other hand, anti-S IgG persisted in most women from 6 months to at least one year after infection, while anti-S IgA and anti-NCP IgG declined before [132].

## 6. Conclusions

In this review, we have summarized the complexity of the immune response against viruses that affect the airways. Despite decades of efforts to reduce the global impact of these infections through vaccination strategies and preventive policies, the recent COVID-19 pandemic has highlighted the ongoing need to understand how viruses can be dangerous, particularly for newborns and pregnant women. Studying the ability of viruses to evade the host defense, the spread of new variants, and identifying weak points in the human immune system are essential issues that need to be explored in future research.

## Figures and Tables

**Figure 1 ijms-25-06178-f001:**
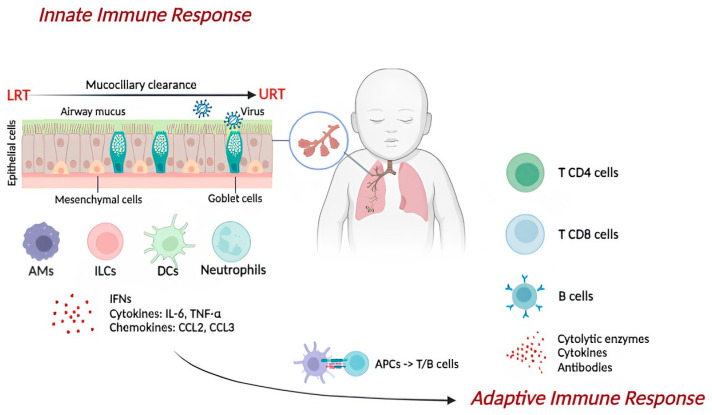
Representation of the immune response during respiratory viral infections. First, the virus activates an innate immune response constituted by structural cells (i.e., epithelial and mesenchymal cells), chemical barriers (mucus produced by goblet cells), and several immune cells (i.e., alveolar macrophages (AMs), innate lymphoid cells (ILCs), dendritic cells (DCs), neutrophils). Interferons (IFNs), cytokines (such as IL-6 and TNF-α), and chemokines (such as CCL2 and CCL3) also promote viral clearance. The adaptive immune response starts in the draining lymph nodes after interacting with specific cells (i.e., T CD4 and T CD8 cells, B cells) and the antigen-presenting cells (APCs).

**Figure 2 ijms-25-06178-f002:**
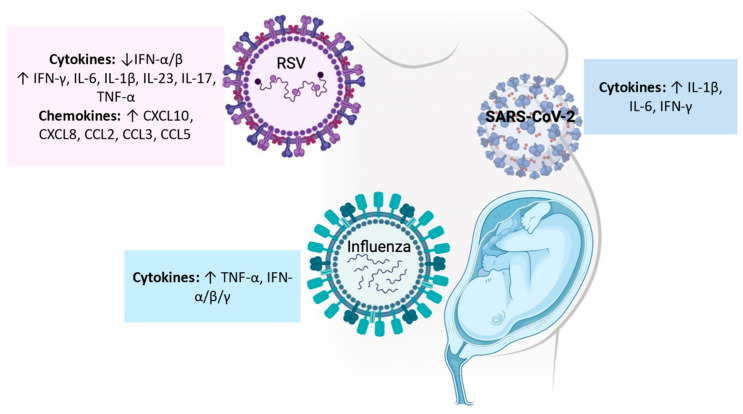
Role of cytokines and chemokines during respiratory viral infections in particular populations (newborns and pregnant women). RSV inhibits IFN-α/β signaling, while it induces upregulation of some cytokines (i.e., IL-6, TNF-α) and chemokines (i.e., CXCL10, CXCL8, CCL2, CCL3, CCL5). During influenza type-1, IFNs, and TNF-α are crucial in limiting viral replication. SARS-CoV-2 infection in newborns was related to high levels of IL-1β, IL-6, and IFN-γ.

**Table 1 ijms-25-06178-t001:** Comparison between principal innate and adaptive immunological mechanisms associated with mechanisms of evasions of RSV, SARS-CoV-2, and influenza virus.

Immunological Mechanism	RSV	SARS-CoV-2	Influenza Virus
** *Innate immune response* **			
PRRs	TLR2, TLR3, TLR4, TLR7, and TLR8 recognize the virus [58]. RIG-I and MDA5 recognize the virus [59]. NLRP3 promotes secretion of pro-inflammatory cytokines [60]. NLRC5 regulates IFN-I expression [58].	TLR2 recognizes the virus, triggering the release of TNF-α and IFN-γ [61]. TLR3 induces the production of IL-1β and IL-18 via the NLRP3 inflammasome [61]. TLR4, TLR1, TLR5, TLR7, TLR8, and TLR9 detect viral RNA, realizing cytokines, including IFN I/III [62]. RIG-I and MDA5 activate NF-kB signaling and IFN I/III [63].	TLR3, TLR7, TLR8, RIG-I, and NLRP3 induce the expression of IFNs I/III and pro-inflammatory cytokines, stimulating antiviral ISGs, and recruit immune cells [64]. NLRP3 inflammasome releases IL-1β and IL-18 triggering pyroptosis in infected cells [64].
IFNs	Viral proteins inhibit IFN production [65]. Increased levels of IFN-λ 1–3 are associated with the disease severity [66].	IFNs I/III activate JAK/STAT pathway and induce the expression of MHC class I and ISGs [67].	IFNs I/III stimulate antiviral ISGs and recruit pro-inflammatory cells [64].
** *Adaptive immune response* **			
T cells’ response	CD4 cells promote the differentiation of cytotoxic CD8 cells and B cells [68]. Th2 response contributes to antibody and eosinophils’ responses [69].	CD4 cells stimulate B cells and activate CD8 cells, which contribute to eliminate virus-infected cells [70].	CD4 T cells produce IFNs and IL-2, provide help to B cells for antibody production, and contribute to the generation and recall of CD8 T cell memory. CD8 cells promote viral clearance and reduce the disease severity [71].
B cells’ response	B cells produce antibodies, and anti-F protein antibodies exhibit superior neutralization capabilities [72]. Neonatal B cells contribute to heightened Th2 response [73]. Reduced IFN responses potentially result in decreased B cell function in newborns [73].	B cells produce neutralizing antibodies [74]. Acute COVID-19 is marked by the absence of germinal centers, leading to the generation of “disease-related” B cells with limited protective capacity [75].	B cells produce antibodies, targeting the surface glycoproteins HA and NA. These antibodies neutralize viral particles, inhibit viral entry and release, and promote opsonization for phagocytosis [76].
** *Mechanisms of evasions* **	NS1 and NS2 suppress IFN-I production and signaling [77]. G, N, M, and SH proteins disrupt innate immune recognition by PRRs and modulate the host’s innate immune response, facilitating persistent infection and recurrent respiratory tract infections [78].	Inhibits the IFNs production and signaling, delaying immune response activation [79]. Evades recognition by TLRs and RLRs and modulates antigen presentation [79]. Manipulates cytokine signaling pathways, exacerbating inflammation and disease severity [79]. Undergoes antigenic variation, evading recognition by pre-existing immunity and leading to reinfection or reduced vaccine efficacy [79].	Rapid mutations of HA and NA allow the virus to escape recognition [80]. Antigenic drift and shift lead to the emergence of novel strains with altered antigenic properties, complicating immune recognition [81]. NS1 inhibits IFN response. Induces immunosuppression, facilitating viral persistence and dissemination [82].

G protein: glycoprotein; HA: hemagglutinin; IL: interleukin; IFNs: interferons; ISGs: interferon-stimulated genes; JAK/STAT: Janus kinase/signal transducers and activators of transcription; M protein: matrix protein; MHC: major histocompatibility complex; MDA5: melanoma differentiation-associated protein 5; N protein: nucleocapsid protein; NA: neuraminidase; NF-kB: nuclear factor kappa-light-chain-enhancer of activated B cells; NLRC5: NOD-like receptor family CARD domain containing 5; NLRP3: NLR family pyrin domain containing 3; NS1-2: nonstructural proteins 1–2; PRRs: pattern recognition receptors; RIG-I: retinoic acid-inducible gene I; RLR: (RIG-I)-like receptors; SH protein: small hydrophobic; Th2: T-helper type 2; TLRs: toll-like receptors; TNF-α: tumor necrosis factor.

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
