# Peer review of "Immune Response to Respiratory Viral Infections"

_ijms, 2024, doi:10.3390/ijms25116178_

Round 1

Reviewer 1 Report

Comments and Suggestions for Authors

Gambadauro and colleagues have written a helpful review article that pulls together a great deal of information on the respiratory immune response to flu, RSV, and SARS-CoV-2.  They've put a special emphasis on immunity in newborns and pregnant women, which I think is an important aspect and strengthens the manuscript.  Overall, I think this manuscript would be quite useful to many people working in this general area.  However, I've noted some opportunities for improvement.

Major Comments

1.       While this is a helpful review article, it relies too heavily on other review articles.  As stated in line 41, “[t]his review aimed to summarize high-quality studies on the immune response to respiratory viral infections.”  This article does not accomplish that task.  Of the first 20 articles listed among the references for this manuscript, approximately half appear to be review articles (references 2, 3, 4, 6, 2, 13, 14, 16, 17, 18, 20).  I understand the convenience of citing other review articles in this review article, but it makes it more difficult for the reader to reference the primary literature.  The article would be improved if the authors would more heavily in cite the primary research literature rather than other review articles.  To be clear, citing some review articles is unavoidable, but I think it should be a minor fraction in any given section of the paper.

Minor Comments

1.       Line 19 – consider revising to “…pathologies must not decrease.”

2.       Figure 1 – the blue non-ciliated cells in the upper left portion of the figure need to be labeled.  Mesenchymal cells and goblet cells are clearly labeled.  Ciliated epithelial cells are obvious.  What are the blue cells?  My guess would be M cells, but those aren’t mentioned anywhere in the article.

3.       Bibliographic information for reference #4 includes on the title and authors. 

Author Response

REVIEWER 1

Point 1: “Gambadauro and colleagues have written a helpful review article that pulls together a great deal of information on the respiratory immune response to flu, RSV, and SARS-CoV-2. They've put a special emphasis on immunity in newborns and pregnant women, which I think is an important aspect and strengthens the manuscript. Overall, I think this manuscript would be quite useful to many people working in this general area. However, I've noted some opportunities for improvement.”

Response: We are grateful to the Reviewer for the constructive comments to our manuscript.

Point 2: “While this is a helpful review article, it relies too heavily on other review articles.  As stated in line 41, “[t]his review aimed to summarize high-quality studies on the immune response to respiratory viral infections.” This article does not accomplish that task. Of the first 20 articles listed among the references for this manuscript, approximately half appear to be review articles (references 2, 3, 4, 6, 2, 13, 14, 16, 17, 18, 20). I understand the convenience of citing other review articles in this review article, but it makes it more difficult for the reader to reference the primary literature. The article would be improved if the authors would more heavily in cite the primary research literature rather than other review articles. To be clear, citing some review articles is unavoidable, but I think it should be a minor fraction in any given section of the paper.”

Response: We implemented the research literature as requested by the Reviewer.

Point 3: “Line 19 – consider revising to “…pathologies must not decrease.”

Response: We modified the sentence as suggested (page 1, line 19).

Point 4: “Figure 1 – the blue non-ciliated cells in the upper left portion of the figure need to be labeled.  Mesenchymal cells and goblet cells are clearly labeled.  Ciliated epithelial cells are obvious. What are the blue cells?  My guess would be M cells, but those aren’t mentioned anywhere in the article.”

Response: We modified Figure 1 (on page 2) by removing the blue non-ciliated cells.

Point 5: “Bibliographic information for reference #4 includes only the title and authors.”

Response: We modified the reference (page 15, lines 658-659).

Reviewer 2 Report

Comments and Suggestions for Authors

The manuscript presents a review of the immune response to respiratory viruses, focusing on RSV, influenza, and SARS-CoV-2. I found the manuscript generally well-organized, thorough, and readable. Some suggestions:

1. IFN is often written INF, please correct.

2. Influenza does not need to be italicized.

3. LRTI is written as LTRI on line 248.

4. I really like Table 1, but it would be more useful if references were included.

5. I think the authors could also discuss how the immune response changes in the elderly for these three infections since that is one of the higher risk groups for all three infections.

Comments on the Quality of English Language

English is fine.

Author Response

REVIEWER 2

Point 1: “The manuscript presents a review of the immune response to respiratory viruses, focusing on RSV, influenza, and SARS-CoV-2. I found the manuscript generally well-organized, thorough, and readable.”

Response: We are grateful to the Reviewer for the constructive comments to our manuscript.

Point 2: “IFN is often written INF, please correct.”

Response: We modified the acronym.

Point 3: “Influenza does not need to be italicized.”

Response: We removed the italics.

Point 4: “LRTI is written as LTRI on line 248.”

Response: We modified the acronym.

Point 5: “I really like Table 1, but it would be more useful if references were included.”

Response: Thanks for your suggestion. We included references in Table 1 (pages 6-7).

Point 6: “I think the authors could also discuss how the immune response changes in the elderly for these three infections since that is one of the higher risk groups for all three infections.”

Response: We appreciated your suggestion, but our narrative review was specifically focused on summarizing the impact of the three viruses on the human immune system, with a specific emphasis on infants and pregnant women. As pediatricians, we chose to restrict our research to our area of expertise.
